# The Clinical Utility and Impact of Next Generation Sequencing in Gynecologic Cancers

**DOI:** 10.3390/cancers14051352

**Published:** 2022-03-07

**Authors:** Vijaya Kadam Maruthi, Mahyar Khazaeli, Devi Jeyachandran, Mohamed Mokhtar Desouki

**Affiliations:** 1Department of Pathology, University at Buffalo, Buffalo, NY 14260, USA; vkadamma@buffalo.edu (V.K.M.); mahyarkh@buffalo.edu (M.K.); 2Department of Pathology, Roswell Park Comprehensive Cancer Center, Buffalo, NY 14203, USA; devi.jeyachandran@roswellpark.org

**Keywords:** molecular testing, gynecologic tumors, next generation sequencing

## Abstract

**Simple Summary:**

Cancer cells harbor many genetic abnormalities, but the key oncogenic pathways that lead to clinically evident cancer require driver mutations termed actionable mutations. These actionable mutations can be detected using genomic profiling or next-generation sequencing tests. This discovery has led to a tremendous change in treatment regimens from standard chemotherapy to targeted therapy where drugs are specifically targeted against these actionable mutations. Due to the cost-effectiveness and various testing platforms, utilization of these tests by oncologists has increased enormously, but the impact of targeted therapy based on these test results is still understudied. We aimed to identify the clinical utility rate of the tests and analyze the survival benefit for those receiving targeted therapy based on the test results of gynecologic cancer patients. Our findings showed high clinical utility of the tests used by gynecologic oncologists along with a significant survival benefit.

**Abstract:**

Next generation sequencing (NGS) has facilitated the identification of molecularly targeted therapies. However, clinical utility is an emerging challenge. Our objective was to identify the clinical utility of NGS testing in gynecologic cancers. A retrospective review of clinico-pathologic data was performed on 299 gynecological cancers where NGS testing had been performed to identify (1) recognition of actionable targets for therapy, (2) whether the therapy changed based on the findings, and (3) the impact on survival. High grade serous carcinoma was the most common tumor (52.5%). The number of genetic alterations ranged from 0 to 25 with a mean of 2.8/case. The most altered genes were *TP53, PIK3CA, BRCA1* and *BRCA2*. Among 299 patients, 100 had actionable alterations (79 received a targeted treatment (Group1), 29 did not receive treatment (Group 2), and there were no actionable alterations in 199 (Group3). The death rate in groups 1, 2 and 3 was 54.4%, 42.8% and 50.2%, with an average survival of 18.6, 6.6 and 10.8 months, respectively (*p* = 0.002). In summary, NGS testing for gynecologic cancers detected 33.4% of actionable alterations with a high clinical action rate. Along with the high clinical utility of NGS, testing also seemed to improve survival for patients who received targeted treatment.

## 1. Introduction

The use of next generation sequencing (NGS), a technique of massively parallel sequencing of millions of fragments of DNA, has been rapidly growing because of its ability to simultaneously analyze several genes or gene regions from a single assay [1]. NGS technology has contributed to a better understanding of the genomic and molecular pathogenesis of malignancies, the discovery of novel biomarkers for screening, and the prevention and significant development of precision medicine [2]. One of the rapidly expanding NGS platforms in clinical practice is the detection of actionable alterations which can be targeted by drug therapy, as NGS has unveiled previously unrecognized genomic alterations which are now amenable to targeted therapy. However, its impact and usefulness are still largely underdetermined despite the growing utility of NGS in real world clinical practice.

We retrieved cases of gynecologic carcinomas (including tumors of primary adnexal, uterine, cervical, and vulvar origin) which underwent NGS testing on formalin-fixed paraffin-embedded (FFPE) tissue. We analyzed the prevalence of gynecologic tumors with actionable alterations, and their clinical utility, along with overall survival at our comprehensive cancer center. The goal of our study was to address whether there was improved survival of patients receiving targeted therapy based on actionable alterations.

## 2. Materials and Methods

Upon approval of the study by the institutional review board, clinico-pathologic data for 299 gynecological cancer patients who underwent NGS testing between 2014 and 2020 were retrieved from our institutional electronic medical records. The patients’ clinical and pathologic data were reviewed encompassing demography, pathology results including tumor histotype, local recurrence and distant metastasis, NGS test results, targeted treatment based on NGS results and follow-up.

At our institution, the NGS testing was requested by the clinicians. More than 90% of the NGS testing was performed in-house (OmniSeq, Buffalo, NY, USA) and the remainder was outsourced (Foundation Medicine Inc., Cambridge, MA, USA). OmniSeq Comprehensive^®^, a commercially available test approved for clinical use by the New York State Clinical Laboratory Evaluation Program (NYS CLEP) [3] was used. The OmniSeq Comprehensive CGP assay involves cDNA sequencing of tumor tissue to identify somatic alterations in 144 cancer-associated genes, including single nucleotide variants, insertions, deletions, indels, copy number variants, and RNA sequencing to perform rearrangement (fusion) analysis in oncogenes (https://www.omniseq.com/omniseq-insight/ accessed on 3 March 2022). The recently developed extensive panel, which includes 522 genes was not the platform used for our patient cohort.

Germline alterations are not tested by the OmniSeq platform. However, germline mutations can be extracted from the somatic tumor sequencing results. As such, the test reports detected mutations in genes defined by the American College of Medical Genetics and Genomics (ACMG) as potentially hereditary and directs the physicians to further investigate by germline testing if clinically applicable.

The endpoint of the study was to assess the overall survival based on targeted therapy based on the NGS results. Overall survival was defined as the time from performing the NGS testing to the last follow-up or death. Kaplan–Meier survival curves for overall survival were analyzed based on the genomic alterations and treatment, and the *p* values were obtained using a log-rank test. A *p*-value < 0.05 was considered to be statistically significant.

## 3. Results

Among 299 gynecologic cancers, common tumor types tested were high grade serous carcinomas (52.5%), endometrioid carcinomas (17.0%), squamous cell carcinomas (5.7%) and carcinosarcomas (5.3%) (Table 1). The sites of the primary tumors were the ovary (50.5%), endometrium (29%), cervix (9%) and the fallopian tube (3.7%) (Table 1). Only one case of primary peritoneal high grade serous carcinoma was included in “other tumors”. The tumor had one genetic alteration at the *TP53* gene. Tumor grades were high, intermediate, and low grade in 73.6%, 6% and 14.7% of cases, respectively. The mean age at the NGS testing was 61 years. The interval between the diagnosis and the NGS testing ranged from 0 to 457 months with an average of 40.4 months. The majority (86.9%) of the patients were diagnosed as stage 4 (68.3%), followed by stage 3 (18. 6%) at the time of testing, and 3.1% of the cases had no staging information. The number of alterations ranged from 0 to 25 with a mean of 2.8 per case (Figure 1). Table 2 summarizes the frequently altered genes in the study population according to tumor type. *ARID1A* was one of the genes included in the 144 gene panel and was seen in four cases; one clear cell carcinoma showed *ARID1A* alterations along with *KDM, PPP2R1A, R183W, M2055, Q515, Q147*. The other three cases were high-grade serous carcinomas that showed *ARIDA1* along with *TP53* alterations.

The common targeted therapies given were olaparib, everolimus, rucaparib, niraparib and pembrolizumab. The most common drug was Olaparib, given as monotherapy ranging from 2 to 6 cycles, or in combination with tremelimumab and durvalumab for one year (clinical trial I288216), in patients who had received three or more lines of chemotherapy previously. It was usually prescribed in our patient population at a dose of 300 mg twice daily until disease progression or unacceptable toxicities, such as severe anemia, severe thrombocytopenia, renal toxicity or fatigue.

We defined actionable alterations as those that could be targeted by a drug (on-label or off-label, or in clinical trial). Among 299 patients, 100 (33.4%) had actionable alterations or genetic alterations for which a targeted therapy was available. Subsequently, 79 (79%) received a targeted treatment (designated in our study as Group 1), 21 (21%) had no change in treatment (designated in our study as Group 2), while NGS results did not show any actionable alterations in 199 patients (designated in our study as Group 3) (Figure 2). 

The clinical follow up in Group 1 ranged from 1–56 months (average of 21.3 months) where 43/79 (54.4%) of the patients died after receiving NGS-based targeted therapy, ranging from 1–48 months with an average of 18.6 months. The clinical follow up in Group 2 ranged from 0–26 (average of 8.2) months and 9/21 non-compliant patients died. NGS-based targeted therapy ranged from 0–23 months with an average of 6.6 months. The clinical follow up in Group 3 ranged from 0–76 months with an average of 14.2 months, and 100/199 patients died within 0–50 months with an average of 10.8 months. The death rates in Groups 1, 2 and 3 were 54.4%, 42.8% and 50.2%, with an average survival of 18.6, 6.6 and 10.8 months, respectively, showing a statistically significant difference (*p* value 0.002) (Figure 3). 

## 4. Discussion

In this era of modern technology, genetic testing has progressed from single-gene based assays to much more complex next-generation sequencing (NGS) based assays. NGS is a rapidly evolving technology that has facilitated fast- and high-throughput evaluation of many genes in a short timeframe. It has provided a platform for comprehensive molecular analysis of gynecologic malignancies and has revealed a wide spectrum of mutations and molecular differences, which can be exploited in different arenas, such as understanding of pathogenesis, accurate diagnosis and also targeted therapy [4,5,6,7]. Currently, NGS testing is largely performed on patients who fail to respond to traditional regimens or disease progression [8,9]. While this technology has been widely implemented, there are no standard recommendations from scientific bodies about its utilization in daily clinical practice. Although the cost of NGS has recently decreased significantly, it depends on several factors, such as whether the testing is performed in an academic center or clinic, insurance coverage, in-house vs. outsourced, and extent of sequencing, among other factors [10].

Among gynecologic malignancies, ovarian cancer (OC) is the second most common and the most common cause of gynecologic cancer mortality in the United States. Poly ADP ribose polymerase (PARP) inhibition of cells containing a defect in homologous recombination pathways (e.g., those with *BRCA1*/2 mutations) results in the death of target tumor cells while sparing normal cells. Recently FDA approved olaparib, a PARP inhibitor, to be given as effective maintenance therapy in patients with platinum-sensitive ovarian cancer who are in complete or partial response following platinum-based chemotherapy [11]. The genetic alterations for high grade serous ovarian carcinoma observed in our study were similar to those found in contemporary studies, with the most common genes involved being *TP53*, *BRCA1* and *BRCA2* [5,12,13,14,15].

In endometrial carcinomas, the frequently altered genes were *PIK3CA* (51%) and *PTEN* (43.1%), consistent with the findings of similar studies [6,16]. In cervical squamous cell carcinomas, the commonly altered genes were *PIK3CA* (35.3%) and *FBXW7* (17.6%), consistent with similar studies [7,17]. We assessed the clinical utility and impact of NGS testing in our institution. NGS testing detected actionable alterations in 33.4% of patients, out of which 79% patients received targeted therapy. Though only one third of the cases showed actionable mutations, a high clinical action rate of 79% was observed. We did find a statistically significant difference in survival for patients who showed actionable alterations with subsequent targeted therapy in comparison with patients who, despite showing actionable alterations, did not receive targeted therapy and patients who did not show any actionable alterations on NGS testing. In comparison, one recent similar study demonstrated an overall clinical action rate of 36% for gynecologic cancers. However, the differences in the survival based on the presence/absence of an actionable target and subsequent management were not statistically significant (*p* = 0.516) which was attributed to the small size of the cohort (73 patients). Despite identification of actionable mutations and subsequent appropriate management, this did not translate into improved survival in this study [18]. 

Gynecologic cancers are known to harbor low rates of actionable mutations, especially in comparison to lung, colon, and breast cancers which frequently express alterations in *EGFR*, *KRAS*, and *HER2*, respectively. Though the utility of NGS testing is gaining popularity in oncology clinical practice, there have been few studies evaluating its impact on treatment response or patient survival. In the literature, the benefits in gynecologic cancer treatment have been viewed as debatable [8,19,20,21]. This could be due to various reasons, including: (a) differences in the rate of utilization of the NGS test depending on whether the test is performed in-house or is outsourced; (b) low frequency of actionable mutations in gynecologic tumors; (c) optimal timing, whether the tumor is newly diagnosed, recurrent or metastatic as patterns of genomic alterations may differ; (d) patient clinical status or how well the patient can tolerate targeted therapy and whether they had been previously heavily treated; (e) broader or narrower sequencing approaches; (f) diversity of tumor types; (g) accessibility of the targetable drugs (often off-label drugs), and (h) intra- and inter-patient heterogeneity, among others. Despite these differences, our study showed a significant survival benefit. Other than identification of targeted therapy, subsequent appropriate management by clinicians also plays an important role, which could have contributed to the improved survival in our study.

There are a number of limitations to our retrospective observational study. We did not focus on tumor mutation burden or genetic counseling based on the NGS results We did not study variants of unknown significance as this data is vast, very dynamic and ever-changing. Finally, we did not address the optimal timing of NGS testing or patient selection criteria for maximum benefit. More studies are required in the future to address these issues associated with NGS testing and its consequences.

## 5. Conclusions

Although the molecular landscape in gynecologic cancers is heterogeneous, our data supports the view that the utility of NGS testing can be clinically translated into significant survival benefit. Even when standard treatment fails, NGS testing provides a promising avenue for gynecologic cancer patients. Larger prospective studies are needed to further evaluate the clinical benefits of NGS testing and to determine clinical applicability in the future.

## Figures and Tables

**Figure 1 cancers-14-01352-f001:**
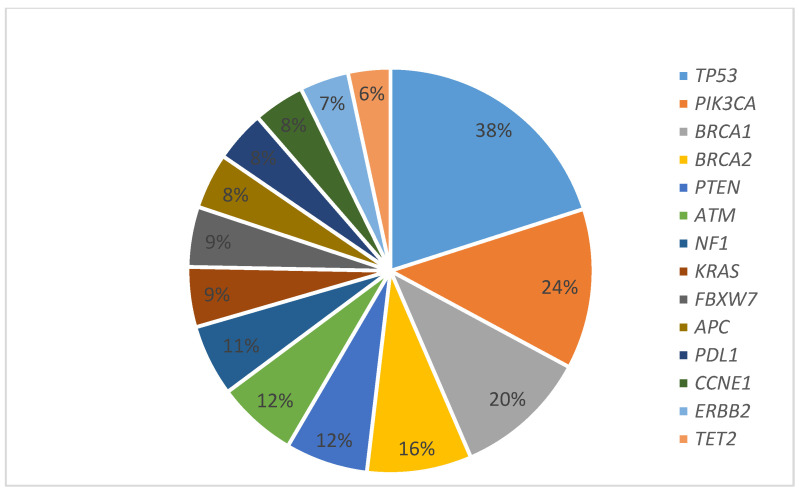
Genes most frequently altered in our cohort (The number of alterations ranged from 0 to 25 with a mean of 2.8 per case).

**Figure 2 cancers-14-01352-f002:**
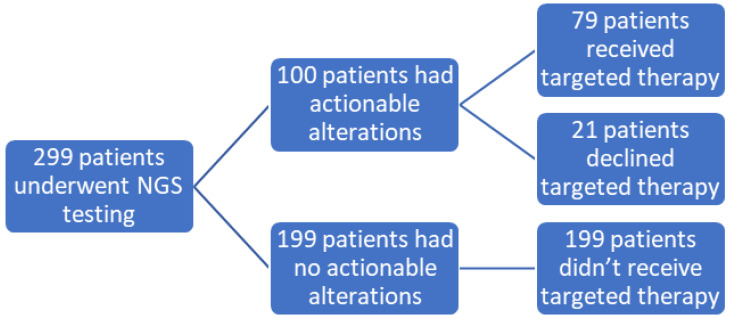
A diagrammatic representation of our study population who received targeted therapy based on NGS testing.

**Figure 3 cancers-14-01352-f003:**
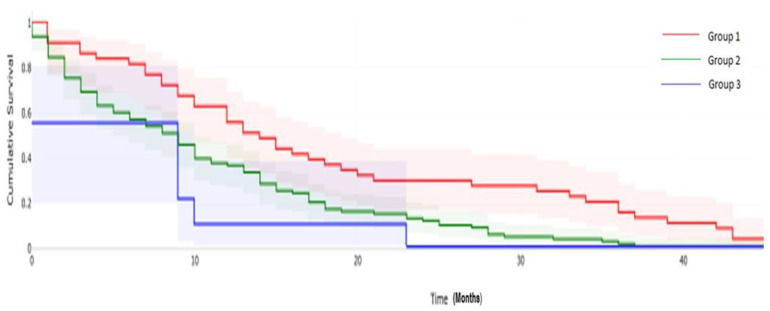
Overall survival based on next-generation sequencing (NGS) test result. *p* = 0.002 by log-rank test.

**Table 1 cancers-14-01352-t001:** Histopathologic characteristics of the cases tested by NGS.

Tumor site	Number	Percent
Ovary	151	50.50%
Endometrium	87	29.10%
Cervix	27	9.03%
Peritoneum	11	3.68%
Fallopian tube	11	3.68%
Adnexa	5	1.67%
Vulva	2	0.67%
Vagina	2	0.67%
Unknown	3	1.00%
Tumor histotype		
High grade serous	157	52.51%
Endometrioid	51	17.06%
Squamous cell carcinoma	17	5.69%
Carcinosarcoma	16	5.35%
Clear cell carcinoma	11	3.68%
Low grade Serous	8	2.68%
Cervical adenocarcinoma	5	1.67%
Adenosarcoma	3	1.00%
Adult granulosa cell tumor	3	1.00%
Mucinous carcinoma	3	1.00%
Borderline Serous	2	0.67%
Dedifferentiated carcinoma	2	0.67%
Sertoli-Leydig cell tumor	2	0.67%
Endometrial stromal sarcoma	1	0.33%
Leiomyosarcoma	1	0.33%
Mucinous, borderline	1	0.33%
Sero-mucinous	1	0.33%
Others	15	5.02%
Pathologic Tumor stage (pT)		
pT1	28	9.60%
pT2	12	4.10%
pT3	54	18.60%
pT4	196	68.30%

**Table 2 cancers-14-01352-t002:** Frequently altered genes in common tumor subtypes of our study population.

HGSC	EC	SCC	Carcinosarcoma	CCC	LGSC
Gene, %	Gene, %	Gene, %	Gene, %	Gene, %	Gene, %
*TP53*, 56.7	*PIK3CA*, 51	*PIK3CA*, 35.3	*TP53*, 56.2	*PIK3CA*, 36.4	*APC*, 25
*BRCA1*, 26.1	*PTEN*, 43.1	*FBXW7*, 17.6	*PIK3CA*, 50	*BRCA1*, 27.3	*BRCA2*, 25
*BRCA2*, 17.2	*ATM*, 23.5	*NFE2L2*, 17.6	*APC*, 25	*BRCA2*, 18.2	*KRAS*, 25
*NF1*, 16.6	*KRAS*, 17.6	*PDL1*, 17.6	*ATM*, 25	*JAK3*, 9.1	*NRAS*, 25
*PIK3CA*, 12.7	*CTNNB*, 17.6	*BIRC3*, 11.6	*PTEN*, 25	*NOTCH1*, 9.1	*CDKN2A*, 25

Abbreviations: HGSC, high grade serous carcinoma; EC, endometrioid carcinoma; SCC, squamous cell carcinoma; CCC, clear cell carcinoma; LGSC, low grade serous carcinoma.

## Data Availability

Not applicable.

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
