# Peer review of "The Clinical Utility and Impact of Next Generation Sequencing in Gynecologic Cancers"

_cancers, 2022, doi:10.3390/cancers14051352_

Round 1

Reviewer 1 Report

Maruthi et al. reported the clinical utility and impact of next generation sequencing in gynecologic cancers. The clinical question is important for precision medicine for gynecologic cancer. The manuscript is well-written and very persuasive. However, some points are inadequate to acceptance. The authors should consider the following points.

1) Germline findings is important for precision medicine of gynecologic cancer from the next generation sequencing. Authors should show the germline findings if authors have the information.

2) Authors describe gene name with italic style.

Reviewer 2 Report

It would have been interesting to have included one of the rarest cancers, so-called “peritoneal cancer” which behaves very much like HGSC. Just thinking about adding a degree of novelty to the manuscript given that the focus of this report is seem countless times in the literature.

Interesting to see that the NGS panel the authors used “somatic alterations in 144 cancer-associated genes” and provide a link to a webpage that doesn’t list the genes interrogated. The webpage goes to in the Insight panel which includes 522 genes. Why didn’t the authors use the more extensive panel. The list of 144 genes must be listed in the Supplementary data. Was ARID1A included in the 144?

Was ARID1A included in the 144 genes? One would have expected to see a percentage of the clear cell cancers and even the endometriod presenting with ARID1a mutations. Of further interest would be the percentage of these cancers which carried mutations in TP53. The authors should provide data on ARID1A or at the very least include a sentence indicating whether or the 4th most mutated oncogene/tumour suppressor wasn’t included in the 144.

Overall, the manuscript is very well written. It would have been useful to provide a graphical representation of the best examples of where NGS and targeted therapy worked well showing the treatment details and PFS over time. Furthermore, the authors do not reference similar studies to compare their success rate and patient uptake of the technology i.e., PMID: 31384390. Otherwise their data fall flat. It is imperative that the authors cite other literature relevant to their work. In summary the Discussion is rather weak.
